# Circular and Circulating DNA in Inflammatory Bowel Disease: From Pathogenesis to Potential Molecular Therapies

**DOI:** 10.3390/cells12151953

**Published:** 2023-07-27

**Authors:** Federica Di Vincenzo, Ylenia Yadid, Valentina Petito, Valeria Emoli, Letizia Masi, Daniela Gerovska, Marcos Jesus Araúzo-Bravo, Antonio Gasbarrini, Birgitte Regenberg, Franco Scaldaferri

**Affiliations:** 1IBD Unit, Centro di Malattie dell’Apparato Digerente (CeMAD), Medicina Interna e Gastroenterologia, Fondazione Policlinico Universitario “A. Gemelli” IRCCS, 00168 Rome, Italy; federica.divincenzo30@gmail.com (F.D.V.); letizia.masi94@gmail.com (L.M.); antonio.gasbarrini@unicatt.it (A.G.); franco.scaldaferri@unicatt.it (F.S.); 2Dipartimento di Medicina e Chirurgia Traslazionale, Università Cattolica del Sacro Cuore, 00168 Rome, Italy; ylenia.yadid@gmail.com (Y.Y.); valeria.emoli@gmail.com (V.E.); 3Computational Biology and Systems Biomedicine, Biodonostia Health Research Institute, Calle Doctor Begiristain s/n, 20014 San Sebastian, Spain; daniela.gerovska@biodonostia.org (D.G.); marcos.arauzo@biodonostia.org (M.J.A.-B.); 4IKERBASQUE, Basque Foundation for Science, Calle María Díaz Harokoa 3, 48013 Bilbao, Spain; 5Department of Cell Biology and Histology, Faculty of Medicine and Nursing, University of Basque Country (UPV/EHU), 48940 Leioa, Spain; 6Section for Ecology and Evolution, Department of Biology, University of Copenhagen, Universitetsparken 13, Room 426, DK-2100 Copenhagen, Denmark; bregenberg@bio.ku.dk

**Keywords:** cell-free nucleic acids, cell-free DNA, circular DNA, microvescicles, inflammatory bowel disease, cGAS-STING, TLR9, oligonucleotides, bioinformatics, molecular therapies

## Abstract

Inflammatory bowel diseases (IBD), including Crohn’s Disease (CD) and Ulcerative Colitis (UC) are chronic multifactorial disorders which affect the gastrointestinal tract with variable extent. Despite extensive research, their etiology and exact pathogenesis are still unknown. Cell-free DNAs (cfDNAs) are defined as any DNA fragments which are free from the origin cell and able to circulate into the bloodstream with or without microvescicles. CfDNAs are now being increasingly studied in different human diseases, like cancer or inflammatory diseases. However, to date it is unclear how IBD etiology is linked to cfDNAs in plasma. Extrachromosomal circular DNA (eccDNA) are non-plasmidic, nuclear, circular and closed DNA molecules found in all eukaryotes tested. CfDNAs appear to play an important role in autoimmune diseases, inflammatory processes, and cancer; recently, interest has also grown in IBD, and their role in the pathogenesis of IBD has been suggested. We now suggest that eccDNAs also play a role in IBD. In this review, we have comprehensively collected available knowledge in literature regarding cfDNA, eccDNA, and structures involving them such as neutrophil extracellular traps and exosomes, and their role in IBD. Finally, we focused on old and novel potential molecular therapies and drug delivery systems, such as nanoparticles, for IBD treatment.

## 1. Introduction

Inflammatory bowel disease (IBD), including Crohn’s Disease (CD) and Ulcerative Colitis (UC) are chronic multifactorial disorders which affect the gastrointestinal tract with a variable extent, typically leading to the development of symptoms, such as rectal bleeding, abdominal pain, diarrhea and weight loss [1]. Although the pathogenesis of IBD remains still unknown, their development is considered as the result of genetic, environmental, gut microbiota and immunological factors [2]. The diagnostic procedures of IBD are time consuming, requiring endoscopy, blood and stool exams, ultrasonography, or magnetic resonance imaging. Sometimes these procedures need to be repeated multiple times before a definitive diagnosis is reached, prolonging the discomfort of the patient [3]. Therapeutic possibilities are limited, with rates of primary or secondary non-response to therapies reaching 50–60% of treated patients [4,5]. Moreover, despite ongoing research efforts, to date there are no biomarkers that can predict clinical response to available drugs, hence the need for continuous disease monitoring of IBD patients with invasive examinations, such as endoscopy.

In this respect, cell-free Nucleic Acids (cfNA) represent a promising field of research. In complement to the direct analysis of genomic material obtained from cells, the possibility of identifying and analyzing cfNAs present in the circulation or other body fluids would allow the identification of asymptomatic individuals at high risk for IBD [6], or could represent a promising new non-invasive biomarker to distinguish patients with active disease from those in remission or from healthy individuals, allowing the tracking of disease onset, progression and remission following therapy, and hopefully also to predict drug response.

The first findings of cfNA date back to 1948, when Mandel and Métais described it in human plasma [7]. However, only twenty years later, the number of studies have grown, probably due to biotechnological and computational biology advances. Since then, their presence was detected in many easily obtainable bodily fluids, such as blood, saliva, stool and urine, assessing their ever-increasing role in new biochemical analysis. Among these, liquid biopsy reached a wide interest, becoming part of the standard of care in many fields [8], such as prenatal care with the so-called “non-invasive prenatal testing” (NIPT) [9,10], and oncology [11,12,13,14,15].

All cfNA are, according to definition, free from origin cell and able to circulate into the blood. Among them, human derived cfNAs can also travel with or without carriers, such as microparticles (MPs) [16,17,18]. Within MPs, exosomes, appear to have regulatory and influential activity in many dysfunctional conditions, like cancer, chronic inflammation (e.g., IBD, arthritis), thrombosis, pregnancy diseases and sickle cell anemia [19,20,21,22], findings that point towards an ever-growing role in immune system regulation. Thanks to increasing basic science-based studies on this topic and to the development of sophisticated biotechnologies, today the clinical management of the above mentioned (and many more) pathologies is drastically changing. The aim of this review is to contribute with our results to provide a basis for a new target of care for chronic inflammatory conditions like IBDs, that exploits the potential of circulating nucleic acid from understanding the pathogenesis of the disease, to the development of novel molecular therapeutics.

## 2. Cell-Free DNAs (cfDNAs)

Nucleic acids are known to be packed into the nucleus or floating into cytosol, but recently another location has been conferred to them: blood circulation and extracellular space. CfNAs can be released into circulation by various biological processes, so to explicate a regulatory role not only into cells they are normally contained by, but also at distance, in either a coding or a non-coding manner [23,24,25]. Endogenous-derived nucleic acids must be distinguished from exogenous, which mostly derive from microbial (specifically bacterial) DNA. Endogenous nucleic acids result from many processes, among which apoptosis/necrosis, NETosis and active release (by microvesicles (MVs)/exosomes) are the most endorsed ones [26,27,28]. These processes generate RNAs, that have long distracted our attention from cfDNA, nuclear (cf-ncDNA) and mitochondrial DNA (cf-mtDNA) in a linear shape. Nucleic acids can appear totally naked, bound to vesicles or in macromolecular structures (virtosomes, nucleosomes, Neutrophil extracellular traps (NETs)) [27]. Another kind of cell-free DNA, the extrachromosomal-circulating-circular DNA (eccDNA, see below), derives from other biogenesis mechanisms, mostly related to genomic instability, DNA damages and hostile cellular environments [11], but also apoptosis [29] and probably necrosis and pyroptosis. Similar to circulating DNA, which can be found either in a particulate (vesicles enclosed) and non-particulate form (naked DNA, nucleosomes), cfDNAs have been described as circulating free or MVs enclosed molecules, able to sustain propagation or even shut down inflammatory stimulus [30,31,32].

Cells undergo cycles of birth and death, on a frequency based on tissue homeostasis. Cell death can occur unexpectedly (necrosis) or can be programmed (apoptosis) and both processes, in the end, contribute to vesicle bound or naked cfDNA release [33]; thus, high-turnover tissues might contribute consistently to cfDNA circulation [34]. Several studies demonstrated that the circulating fraction of total cell-free DNA varies from 0.1 to 89% [35] suggesting that DNA circulation may occur on specific conditions. An unnatural high-turnover status is surely that of cancer; indeed, many studies associated cancer to abnormal cfDNA levels in serum [11,36]. Evaluations of length, patterns of methylation, nucleosomes alterations and fragmentomics revealed a ‘tissue-of-origin signature’ on cfDNA, proposing that most of it comes from mitotic cells, like those of cancer [37]. Moreover, stability and half-life of cfDNA depend on protein association: long naked molecules (>10 kb), like free DNA fragments released during apoptosis, are rapidly cleaved by DNAse and lost. Instead, shorter molecules (<100 bp) are bound by nucleosomes, protecting them from nucleases and increasing half-life [38], meaning that also the clearance process contributes to DNA plasma levels. Hence, the increased plasma levels of cfDNA observed in IBD may be partly explained by the lower DNAse activity observed in patients with Crohn’s disease [39].

### 2.1. NETs

NETosis is the process by which neutrophils exert their ‘license to kill’: the cell ‘sacrifices’ itself, releasing complex structures of genetic material that entraps mainly bacteria contributing to the cfDNA pool. It consists in nuclear or mitochondrial oxidized DNA, histones and protases release, which have pro-inflammatory capacities [40,41]. NETs also contain cathepsin-G and calprotectin, a well-known biomarker in IBD [42,43]. Immunogenicity of NETs consists primarily in stimulating many cell activities: activation of TLR of dendritic cells (resulting in IFN-alpha synthesis), Immunoglobulin (Ig) class switching in B cells and T cell response boosting [44]. It should be reminded that neutrophils can randomly circulate and be recruited from circulation to specific sites. This means that NETs can be released directly into circulation or into another tissue whose venous drainage can return NETs components to systemic circulation or degrade them directly in situ, without passing to blood. Thus, technically, not all of NETs released into the organism contribute to the circulating DNA pool [45]. An IBD gut is known to undergo a sustained and chronic inflammation process, where both unbalanced T cell priming and microbial dysbiosis are features. Although it is not clear who first started the inflammatory processes, if NETs are the “primum movens” or the answer to a preexistent noxa, it is increasingly clear that both microbiota alterations and microenvironment participate to NETosis [26] and so to mucosal damage. Specifically, in IBD, NETs seem to have pro-thrombotic abilities, through direct platelet activation [46] but probably also by contributing to endothelial damage at blood-cell interface. However, NETs can regulate many cytokines secretion during inflammation, and manage the clearance of damage-associated-molecular-patterns (DAMPs) in mice [47]. Recently, this mode has been extended also to eosinophils, cells able to release eosinophil extracellular traps (EETs), which apparently correlate with disease severity [48]. Ultimately, NETs, along with calprotectin, do correlate with inflammation severity [28,49], but due to their multiple functions, they are not an optimal target as a therapy or as markers in IBD.

### 2.2. Vesicle-Bound DNA

Extracellular vesicles (EVs) or microparticles are cell-released membrane covered globules that include microvescicles (microparticles in the literature), exosomes and others [50], and can differ in size, composition, role and genesis processes [51]. EVs are released from almost all cell types, including bacteria (Bacterial Extracellular vesicles, BEVs) [50,52], under specific circumstances. Their presence has been reported in saliva [53], sperm [54], milk [55], urine, blood (serum and/or plasma) [56] among others. Often, isolated vesicles were attributed to exosomes, since their exosome-like protein cargo; however, circulating vesicles are probably both MVs and exosomes. Indeed, studies revealed that luminal EVs in IBD patients are less than 500 nm in diameter [27,31], confirming the presence of both exosomes and microvesicles. In the last decade, increasing interest has arisen, since it has been demonstrated that vesicles shed from plasma membranes of cells in complete physiological conditions, and carry a significant amount of interesting biological material [57]. Many pathological conditions started to benefit from MPs role, not just as possible biomarkers; EVs are elevated in pneumological and rheumatic diseases, while urinary EVs seem to reflect acute kidney injury [58]. Notably, their peculiar structure inspired new drug delivery systems [59].

Exosomes are considered as single-layer lipid membrane vesicles of 30–150 nm diameter [27,31]. Cellular trafficking generates exosomes, born from the so-called multi vesicular body (MVB), that can become a late endosome, from which exosomes derive, or fuse to lysosomes and be degraded [60], while macrovesicles derive from direct outward blebbing from plasma membrane [57]; both are known to mediate intercellular communication. Exosomes can be considered a frozen image of cell conditions, since they reflect its content [61].

In EVs originating from dendritic cells (DCs), besides MHC-I and MHC-II, CD86 costimulatory factor is present [15]; specific integrins on EVs surface are a cell-specific signature. IECs derived exosomes contain β-defensines, antimicrobial molecules immunoglobulins and heat shock proteins (HSPs) [15]. HSPs, a common finding also in DCs exosomes, are known to bind TLR2 and 4, as both Gram positive and Gram negative bacteria do, leading to proinflammatory signaling [62]. In contrast, exosomes deriving from granulocytes myeloid-derived suppressor cells are able to inhibit Th1 and induce Treg proliferation, modulating inflammation [63]. Released exosomes are able to bind surface proteins of other cells or be internalized, in order to exert intercellular communication [22].

Although our focus is on DNA, proteins found in exosomes play a fundamental modulatory effect on immune cells, thus inseparable from inflammation pathogenesis. So far, in IBD patients, exosomes proteins are known to be involved in (I) immunity regulation: [32], (II) regulation of intestinal barrier [64] (III); regulation of intestinal microbiota. Indeed, in pediatric IBD patients, vesicles from mucosa-luminal interface show altered proteome (ROS, MPO loaded vesicles), that apparently correlates with microbiota modifications and increase in microbiota defensive systems, features of aberrant host-microbiota interactions that finally lead to worsened disease activity [65].

Haisheng Liu and coworkers [27] analyzed EV-DNA through nano-flow-cytometry at a single vesicle level from human colorectal cancer cell line (HCT-15) and normal human colon fibroblast cell line (CCD-18Co), adding precious information to the currently limited knowledge on the topic. With the results achieved, it becomes clearer that cfDNA can be contained either on surface and inside a vesicle, in a double-strand shape (ds-cfDNA) or single stranded (ss-cfDNA). Length oscillates from 200 bp to 5000 and no histones are present; thus, surface EV-cfDNA is labile to DNase activity. Conversely, inner EV-cfDNA is protected by EVs envelope and so it is very stable. Moreover, EVs exist in two size peaks: large EVs, that might be called microvescicles, of 80–200 nm diameter, containing less and smaller DNA fragments (0.2–2 Kbp), and small EVs (<100 nm), called exosomes, that carry longer and more DNA fragments.

## 3. EccDNA

eccDNA is a non-plasmidic, circular and closed nucleic acid macromolecule of different dimensions. It resides mainly in nuclei of cells, but has also been found in plasma as circulating eccDNA [11] and it has so far been detected in all the species and tissues tested from mammals, birds, insects, plants and yeast [12,66,67,68]. The dimensions vary from hundred bp to mega bp in cellular eccDNA and the elements are thereby large enough to carry entire genes that are at least occasionally expressed [14]. Extracellular eccDNA in plasma appears to be smaller and follow a periodic pattern of around 200 bp, 400 bp and 600 bp. This has led to the suggestion that eccDNA in plasma derives form DNA that has been wrapping around one or more nucleosomes, and that apoptosis is the likely cause of these eccDNA [11].

### Biogenesis of eccDNA in Tissue

How eccDNA forms in human cells is still largely unknown. Studies of cancer and cancer cell lines suggest that eccDNA formation is associated with DNA damage and repair [69]. A number of repair pathways have been suggested to be involved in the formation of eccDNA including non-homologous end joining (NHEJ) [69,70,71]. If homology occurs between termini, homology-dependent repair pathways will be involved (micro homology mediated end joining, homology recombination, mismatch repair) [69,72]. The formation will also likely depend on the repair pathways expressed in the given tissue and whether the eccDNA arises from single stranded or double DNA breaks, as suggested in [73].

Processes related to DNA instability can also lead to large eccDNA. This is the case for cells that have undergone chromothripsis, which consists in a single catastrophic event of chromosomes which break at multiple points [72]. Chromotripsis is found to occur in 2–3% of human cancers, but whether this and other types of DNA instabilities play a role in IBD is unknown.

For a better description of eccDNA classification, biogenesis, evolution and functions, the reader is directed towards the recent reviews on the subject [73,74].

## 4. CfDNA and eccDNA in Inflammation

As described above, cfDNA are found either associated to MPs or free. Both display influential activity on the two immune system branches, with particular reference to the innate immune cells, as it follows.

Innate immunity is a branch of immune system able to detect pathogen derived ‘non self’ molecules, known as pathogen-associated molecular patterns (PAMPs), through pattern recognition receptors (PRRs), as Toll-like receptors (TLRs), NOD-like receptors (NLRs) (Janeway, 1989) [75], explaining pathogen-mediated inflammation. PAMPs free-inflammation, on the other hand, was better understood when Polly Matzinger [76] proposed the ‘danger’ theory, claiming that molecules that reflect cellular health, released during cellular distress or tissue damage, called damage associated molecular patterns (DAMPs), can induce activation of the same immune cells, leading to ‘sterile-inflammation’. Since then, many molecules, including cfNAs have been identified as DAMPs. Specifically, both DAMPs and PAMPs are ligands of TLR, that are known to be located either on the plasmatic and endosomal membrane [77]. Among TLRs that are known to be nucleic acids specific [77], TLR9 is able to sense foreign (pathogens, mainly bacteria) and host DNA motifs, such as CpG islets [78] and cause downstream NF-kB signaling.

cfDNA-binding molecules, such as histones, are the ligands of TLR2 and 4, whose activation results in the production of TNF-α, IL-6, IL-10 and MPO [78,79,80]. Moreover, histones are able to elicit NETosis, hesitating in more histones release [81] and so propagating inflammation in a loop; notably, core histones act as auto-antigens in systemic lupus erythematosus (SLE) [82]. Nucleosomes, on the other hand, exert stimulation of different inflammatory pathways that still lead to cytokine secretion [83,84]. High-mobility group box1(HMGB1)-DNA complexes bind RAGE (receptor for advanced glycation end products) and are carried by early endosomes for TLR9 recognition, causing activation of DCs and B cells [85].

Almost all nuclear components are massively exposed to extracellular environment or to blood stream during cell death and NETosis, displaying their immunogenicity especially in rheumatic disease, where ANAs abundantly circulate and eventually bind naked cfDNA, EV-cfDNA, forming immune complexes [20], thus propagating autoimmunity-related complications.

Both nc and mtDNA are able to initiate cyclic guanosine monophosphate (GMP)–adenosine monophosphate (AMP) synthase (cGAS)-stimulator of interferon protein (STING), STING-NF-kB or protein absent in melanoma 2 (AIM2) cascade, either from direct binding to cell surface via MVs or leaking into cytosol, being sensed by genes stimulators [86,87]. Mitochondrial DNA, found in MVs or naked, is not protected by histones and so is smaller than ncDNA (30–80 bp) [88]. Still, it is able to initiate innate immunity cells activation via TLR9 [37,89,90], because of its particular similarity to bacterial DNA, or it can mediate communication between adaptive immunity cells, like T-lymphocytes and dendritic cells [91]. Coherently with their role as DAMPs, cfDNAs (both exo and endogenous) are able to induce either primary and secondary hemostasis, thus being potentially involved in threatening conditions such as DIC (disseminated intravascular coagulation) [46,92].

As previously explained, many types of circulating nucleic acids are described across literature [93,94], but specifically circulating circular DNA is a less explored domain. Nonetheless, the few existing works already helped to gain a better insight. Wang et al. with their latest work highlighted concepts of great importance. They demonstrated that eccDNA is circular genetic fragment deriving from randomly chosen genome sequences, partly generated by cells undergoing apoptosis, that acts as a full-fledged DAMP, able to induce activation of innate immunity (in vitro DCs), throughout a cGAS-STING and TLR9 fashion, that results in INF type I, cytokines and chemokines production [29].

Obermeier et al. showed that bacterial cfDNA rich in CpG motif activate TLR9, worsening the course of DSS-induced colitis [95]. CpG motifs are typical of bacterial free DNA, but just recently attributed either to human cfDNA [78]; in particular, the above mentioned micro eccDNA is an example. This demonstrates how endogenous and exogenous cfDNA share similarities, and that can both activate immune response in a sequence-dependent manner; however, Li et al. in 2012 [80] added novelty to this model, demonstrating what Wang’s team later confirmed: eccDNA potency is to be attributed to its circular shape, but not sequence, since synthetic circular DNA triggered almost the same response and, above all else, that its linear counterpart was not nearly as potent as eccDNA [29]. Specifically, it was proposed that DNA curvature, induced by high-mobility group box 1 proteins (HMGB1) and histones H2A, H2B, significantly enhanced TLR9 binding, in a stereo-specific less than sequence-dependent mechanism. Finally, these studies suggest that self-derived circulating DNA (both non-circular cfDNA in nucleosomes or naked, and eccDNA) are able to induce and sustain the inflammation machinery, especially the ‘sterile inflammation’, which is typical of autoimmune disease, where they actually seem to be involved [82,96].

Conversely, other studies propose that preconditioning with methylated and unmethylated genomic DNA isolated from a probiotic mixture (VSL#3), followed by DSS colitis induction in mice, have anti-inflammatory effect, in a TLR9-dependent way, compared to the control (TLR9 ko mice), that did not show any effect [97]. Likewise, Műzes et al. revealed that pretreatment with a single intravenous injection of colitic cfDNA in DSS-induced colitis mice showed increased TLR9-macroautophagy response and upregulation of Baclin1 expression in the colon, with a decreased disease activity as compared to normal cfDNA injection [98]. These findings confirm the suspicion that cfDNA exerts different behaviors depending on its origin (distressed cells or normal) and on the features of the local immunobiological milieu (inflamed or normal).

As previously mentioned, autophagy can occur following the activation of TLR9 [99]. Since DAMPs can activate TLR9, especially via CpG oligonucleotides [99], it can be stated that cfDNA sensing and autophagy are related: injection of cfDNA triggered increase in TLR9 mediated autophagy, that apparently supported cellular fitness within an inflamed environment, reasonably explaining its protective effect [100]. However, the origin of such cfDNA and mechanisms underlying the anti-inflammatory response are still being elucidated.

In 2021, Zhao et al. demonstrated that levels of EVs containing ncDNA, mtDNA and proteins increased in both plasma and colon lavage from DSS induced active colitis in mice, and in IBD patients, and that these EVs were secreted by IECs [101]. Macrophages cultures enriched in EVs from IBD patients showed inflammatory phenotype and activation of STING pathway, leading to cytokines synthesis. Further analysis showed that dsDNA is necessary to STING activation [31,101]. EV-cfDNA levels adequately correlated with DAI and CDAI (disease activity index, Crohn disease activity index), suggesting that inflammation and cell damage trigger the release of EVs and vice versa, and allowing to definitely confirm that EV-cfDNA could be an activity marker. These results propose that either EV-cfDNA and cf-circulating DNA [28] are probably involved in immune stimulation to develop colitis in both mice and human, throughout different mechanisms but reaching the same result.

Much evidence suggests that EVs carrying nuclear or cytosolic auto-antigens, generated during apoptotic processes [102], undergo biochemical modification that lead to formation of modified self-molecules that break the self-tolerance [103] and trigger the autoimmune response. Post-translational biochemical modification, such as citrullination, can occur in several cellular processes including apoptosis, autophagy and NETosis, contributing to auto-antigen formation in autoimmune disease [102,104]. This model has been approved for Sjögren syndrome, SLE, rheumatoid arthritis (RA), type-1 diabetes mellitus (T1DM) [104,105]. Specifically, citrullinated proteins are found in synovial fluid EVs from RA patients, neutrophils deriving from RA patients show citrullinated vimentin, that is a known autoantibody target in RA, and citrullinated histones [106]. These evidence suggest that EVs derived or DNA bound citrullinated proteins, the so-called “citrullinome” correlate to inflammation severity in the above-mentioned chronic inflammatory disease. Coming to IBD: higher expression of citrullination peptides have been found in colonic biopsies of IBD patients, confirming correlation between citrullination and inflammation [107]; however, these results are not significant and further analyses are needed. Moreover, preliminary analysis by M.-L. Liu et al. revealed that tobacco smoke extract-treated neutrophils release MVs with an inhibitory effect of macrophage phagocytosis [108,109]. Hence, high apoptotic rate and NETosis, characteristic of autoimmune disease (AID) and chronic inflammatory diseases, correlate with increased MVs release and clearance slowdown, resulting in accumulation of apoptotic cells derived auto-antigens and immune complexes, worsening SLE and cutaneous LE [108]. In conclusion, it is clear how MVs and an unbalanced, non-self-resolving immune system can truly mediate modification of local and distant microenvironments, concurring to pathogenesis and progression of AID.

Overall, the above-mentioned studies reveal that in IBD microenvironment, damage is mainly directed to colonocytes and enterocytes, which have been demonstrated to maximally contribute to cfDNA levels in situ and in plasma of induced colitis. Disruption of intestinal epithelial cells (IECs), lining the surface of intestinal mucosa, results in a leaky gut, a way that may increase translocation of exogenous and microbial cfDNAs to circulation. In vivo studies of induced colitis in mice suggest that probably all of those mechanisms of cell stress could act in concert to increase cfDNA plasma levels over time with disease progression [28]. What seems clear is that IECs, immune system and microbiota fit together into this loop, in which cfDNAs play a dominant role across all the disease progression over time, as Maronek et al. reviewed [28], which would imply that eccDNA in circulation has a similar role in disease progression. Consequently, the amount of cfDNA could correlate with DAI and be used as a diagnostic and monitoring tool instead of invasive examinations to assess disease severity. Previous studies regarding the potential diagnostic or prognostic role of different types of cfDNA in IBD are presented in Table 1. Finally, previous studies stressed how damaged or even normal genetic material can truly affect immune activity; however, although propagation mechanisms are ever clearer, initiation ones are not. That is the reason why IBD are still tricky and obscure conditions to heal.

## 5. The Role of Computational Biology in Profiling of eccDNA for Personalized Medicine

The bioinformatics tools for analysis of linear cf DNA for the early diagnosis of diseases, prognosis assessment, and efficacy monitoring, has been widely reviewed [112]. Cell-free eccDNA has a potential clinical application as a biomarker in IBD. To use cf-eccDNA as such, one needs methods that differentiate disease from healthy controls with high degree of specificity. Most of the studies have focused on discovering markers based on global features of the eccDNA, such as difference in their length for different length range distributions [113] or number of detected eccDNA between control and treatment conditions. Chen et al. (2022) found that eccDNAs had aggregation and smaller length in CRC tissues compared to tumor-adjacent tissues [114]. Jiang et al. (2023) observed the eccDNA abundance in gastric cancer tissues was aberrantly higher than that of normal adjacent tissues [115]. Such global differences lack specificity, on one side, in distinguishing between diseases, and sensitivity, on the other side, to distinguish between sample groups, and this problem is even more exacerbated in the case of using plasma samples. To gain more insight into the sequence content of cell-free eccDNA, computational biology methods for differential analysis of eccDNA have been developed such as DifCir [116], based on quantifying the number of eccDNA produced per gene (PpGCs) and finding the statistically significant different PpGCs (DPpGCs). Such more sensitive circulomics method has allowed for detecting differences between sedentary and active skeletal muscle tissue of aged males, a case when neither the transcriptomics methods nor the circulomics methods, based on measuring the abundance and frequency distribution of lengths of the eccDNA, were sensitive enough [116].

The most useful eccDNA biomarkers for personalized medicine are the ones predicted from easily extracted human samples such as plasma. However, plasma collects multiple types of DNA species from all of the tissues of the human body. Thus, it is potentially difficult to discriminate disease specific biomarkers from the blood torrent of DNA fragments coming from tissues that have not necessarily been affected by the disease of the patient. These problems make the prediction of eccDNA markers from plasma samples a daunting task. However, the use of DifCir on the circulomics analysis of cell-free eccDNAs from Systemic Lupus Erythematosus (SLE) patients with DNASE1L3·deficiency have disclosed a distinctive and specific genic eccDNA profile of these patients [117] compared to that of healthy controls, and showed the strong potential of cf-eccDNA in diagnosis of immune-mediated diseases.

Another challenge of eccDNA profiles is that the data do not necessarily follow a Gaussian distribution, which is also seen in human samples that have the intrinsic high variability. Therefore, the standard *t*-test for prediction of statistically significant molecules between two conditions is not applicable. Alternatively, the Wilcoxon test can be applied to determine if medians are different between groups [118], or a “democratic” method has been developed to search for commonly abundant molecules in different samples under less restrictive statistical conditions. The method has been initially reported for transcriptomics data analysis [119], then for DNA methylomics [120] and finally for circulomics data analysis [116] where it detects the common PpGCs (CPpGCs) shared by replicates.

## 6. Towards Molecular Therapies for IBD

### 6.1. TLR9 Therapeutics

Toll-like receptor-9 (TLR9), belonging to the class of Toll-like receptors (TLRs), represent the first line of defense against bacterial pathogens, through the recognition of pathogen-associated-molecular patterns (PAMPs) [121]. The activation of TLRs is typically associated with an inflammatory response that clears the invading pathogens. However, TLRs are also involved in the recognition of the same conserved molecular patterns found in the resident microflora; the commensal recognition of the gut bacteria via TLRs is necessary to tolerate the resident microflora and to maintain the intestinal homeostasis [122,123,124].

Previous studies demonstrated that bacterial DNA or its synthetic oligodeoxynucleotides (ODNs) analogues (immunostimulatory sequence (ISS) ODNs or CpG-ODNs), recognized mainly by TLR9, ameliorated the severity of colitis in murine models [125,126,127]. Particularly, Bleich et al. showed that CpG-ODNs administration in germ-free mice reduced intestinal inflammation, as indicated by histology, decreased proinflammatory cytokines, and increased IL-10 secretion, even without pre-existence of bacterial microflora, thus suggesting that CpG-ODN-induced regulatory T cells are not bacterial antigen specific [125]. Accordingly, Lee et al. showed that the protective effects of probiotics are mainly mediated by their own DNA, rather than by other metabolites or antigens, through the activation of the TLR9 pathway and the production of type 1 Interferon [128].

Rachmilewitz et al., in 2006, revealed that certain classes of CpG-ODNs inhibited, by triggering TLR9, the enhanced production of TNF-α and IL-1beta, ex vivo, by inflamed colonic mucosa of patients with active UC [129].

Furthermore, mouse models deficient in TLR9 are more susceptible to the development of colitis [123], and genetic polymorphisms of TLR9 are associated with and increased risk of IBD in humans [130,131], enhancing the critical role of bacterial DNA sensing in the development of IBD. CpG-ODNs can also elicit an antiapoptotic effect through the TLR9-induced upregulation of heat shock proteins, which usually can protect the gut epithelium against intestinal epithelial barrier dysfunction [132,133,134].

O’Hara et al., in 2012, revealed that infection by Campylobacter jejuni reduces the expression of apical TLR9 in intestinal epithelial cells, thereby disrupting TLR9-induced reinforcement of the intestinal epithelial barrier, and that mice previously exposed to Campylobacter jejuni develop a more severe colitis after low doses of DSS administration, with a significant reduction in levels of the anti-inflammatory cytokine IL-25 and an increase in IL-17, which has an ambiguous role in the pathogenesis of IBD [135].

Based on this evidence, the first synthetic single-strand DNA-based immunomodulatory sequence 0150 (DIMS0150) (Kappaproct^®^, cobitolimod) was developed for the treatment of severe, chronic active, treatment-refractory UC. It contains unmethylated CpG motif and functions as an immunomodulator, inducing the activation of TLR9 pathway in effector T and B lymphocytes, dendritic cells and macrophages [136].

Activation of TLR9 by DIMS0150 results in the induction of regulatory T cells, and in the production of cytokines, such as IL-10 and type I interferons from human peripheral blood mononuclear cells (PBMCs), and seems to increase steroid sensitivity in steroid-resistant UC patients [137]. Kuznetsov et al. identified three potential biomarkers CD168, TSP-1 and IL-1RII, whose response to steroids is enhanced after administration of DIMS0150 [138].

In a pilot study of 2013, DIMS0150 had been administered topically through colonoscopy in eight patients with chronic active UC, elected for colectomy. Respectively, 82% and 72% of all patients demonstrated a clinical response and remission at week 12, and all patients except one avoided colectomy at 2 years follow-up [139].

Subsequently, Atreya et al., in a randomized controlled clinical trial involving 131 patients, demonstrated that dual topical administration of DIMS0150 with colonoscopy is a promising and well-tolerated therapeutic option for UC patients, showing statistical significant improvements in clinical remission with mucosal healing and symptomatic remission at week 4, although the study did not achieve the defined primary endpoint of induction of clinical remission at week 12 in active moderate-to-severe UC patients [140]. However, a clinical trial involving 104 patients with moderate-to-severe UC showed a significant higher achievement of symptomatic remission at week 12 in patients treated with dual topical administration of cobitolimod [141].

In a dose ranging, double-blinded phase IIb study (CONDUCT) involving 213 patients with moderate to severe UC, clinical remission at week 6 was achieved in the cobitolimod 250 mg group at 21% vs. 7% in the placebo group [142].

Interestingly, even the oral administration of a TLR9 modulator (BL-7040) appeared effective, safe and well tolerated in a small cohort of patients with moderately active UC [143]. Results suggest that the TLR9 agonist ameliorates UC by inducing IL-10 and FoxP3 production, IL-17A, IL-17F decrease, recruiting wound healing macrophages and regulatory T cells [122]. Coherently with damage-associated cfDNA biogenesis, mucosal healing may be followed by cfDNA decrease in blood and in situ. Hence, it cannot be excluded that specific cfDNA subtypes could be used as predictive biomarkers of clinical response to this innovative treatment.

### 6.2. cGAS-STING Therapeutics

Autophagy is a highly conserved intracellular process of degradation used to eliminate damaged protein aggregates, organelles, as well as invading pathogens [144]. Previous studies demonstrated that cGAS-STING signaling pathway is not only needed for the activation of autophagy and consequent innate immunity, but conversely it is also subject to negative regulation by autophagy components [145,146].

Indeed, the cGAS-STING signaling pathway is involved in a non-canonical autophagy response requiring only selective autophagy machinery components, including the complex ATG5-12-16L1 and the PIP3P effector WIPI2 [147,148,149,150,151]. Autophagy mediated by STING restricts viral propagation, working as a major host defense program. Autophagy components exert feedback on the regulation of STING activity [147,148,149,150,151]. Indeed, cells deficient in autophagy proteins or treated with drugs that inhibit lysosomal acidification, present increased type I interferon production [152]. The autophagy program also prevents the activation of cGAS-STING signaling pathway by delivering cytosolic DNA molecules toward lysosomes where they are degraded by DNase II [147].

Previous studies revealed an increased expression of STING proteins in murine models of DSS-induced colitis. Further investigation showed that the administration of STING agonists in DSS-induced colitis wild-type mice greatly exacerbated colitis, whereas the severity of colitis was markedly reduced in STING knockout (KO) mice [153].

Accordingly, TMEM173, the gene coding STING, was hypomethylated in the intestinal epithelium of 66 pediatric IBD patients, compared to age- and sex-matched non-inflammatory controls [154]. While hypermethylation of TMEM173 is associated with decreased expression of STING, hypomethylation could cause STING overexpression [155]. Zhao et al. demonstrated that both in murine models of colonic inflammation and patients affected by CD, active colitis was associated with an increased release of EVs containing cf-dsDNA, which, in turn, raised intestinal inflammation in macrophages via activating STING-pathway. The effect disappeared after the removal of exosomal dsDNA, and these findings were further confirmed by authors in STING-deficient mice and macrophages [101].

However, a study by Canesso et al. suggested a protective effect of cGAS-STING-IFN I axis in intestinal inflammation. Indeed, they showed that STING-deficient mice were more susceptible to DSS-induced colitis, enteric infection, especially by Salmonella typhimurium and to T cell-induced colitis. These conflicting results could come from the different experimental design: STING-deficient mice and wild-type were cohoused for 4 weeks before the induction of colitis, therefore cohabitation could result in microbiota transfer, affecting the phenotype of mutant mice [156].

Notably, several other studies support the concept that cGAS–STING signaling pathway contributes to the intestinal inflammation in IBD.

Martin et al. showed that STING-deficient mice suffered from less severe DSS-induced colitis, while the use of STING agonist exacerbated DSS-induced colonic damage and inflammation in wild-type mice [153].

Aden et al. revealed that in intestinal organoids deficient of the gene ATG16L1, a gene involved in autophagy and regulation of endoplasmic reticulum function, there is an augmented activation of cGAS-STING signaling pathway via IL-22. In addition, IL-22-mediated activation of type I interferon signaling was associated with the severity of intestinal inflammation in mice [157].

Recently, Chen et al. demonstrated, in two different works, that the cGAS-STING signaling pathway was statistically activated in the intestine of patients with UC and mouse models of DSS-induced colitis, while levels of Atrial Natriuretin Peptide (ANP) and its receptor were decreased. Moreover, they showed that treatment with ANP attenuated DSS-induced colitis in murine models and repaired gut barrier through the inhibition of STING pathway phosphorylation in colonic tissue and epithelial cells [158,159].

Furthermore, a recent study demonstrated that the exacerbation of experimental colitis following the deficiency of pyroptosis executioner Gasdermin D (GSDMD) depends on the hyperactivation of cGAS-STING pathway. Indeed, GSDMD functions in macrophages are a negative regulator of cGAS-STING-dependent inflammation, thereby protecting against colitis. Accordingly, the administration of cGAS inhibitor RU.521 reduced weight loss, colon shortening, DAI score and histopathological findings in wild type murine models of DSS-induced colitis, totally rescuing the colitogenic phenotype in GSDMD-deficient mice. Treatment with RU.521 can also reduce cGAMP levels and decrease STING, TBK1, and IRF3 phosphorylation during colitis, confirming the potential pathogenic role of cGAS–STING signaling pathway in colitis [160].

Notably, a study by Ahn et al. demonstrated that enterocolitis, exhibited through the loss of IL-10, was completely abrogated in the absence of STING. Indeed, cGAS-STING signaling pathway, stimulated by commensal bacteria, is involved in the maintenance of intestinal homeostasis, through the production in mononuclear phagocytes of both pro-inflammatory and anti-inflammatory cytokines, such as IL-10. Accordingly, it has been reported that in homeostatic conditions, the gut microbiota release DNA-containing membrane vesicles, thereby mediating systemic priming of cGAS-STING-INF I-axis and protecting distant organs against infections in a state of constant preparedness [161]. However, cGAS-deficient mice exhibited less severe intestinal inflammation than STING-deficient mice, possibly suggesting a role for cyclic dinucleotides indirectly regulating STING signaling. The authors also revealed that IL-22 mRNA levels were remarkably increased in the colon of IL-10-deficient mice, suggesting possible negative feedback of IL-10 on IL-22, involving STING [162].

Given the role played by the hyperactivation of the cGAS-STING signaling pathway in the maintenance of intestinal homeostasis and controlling intestinal inflammation, its inhibition could represent a valid therapeutic intervention in IBD. The first selective cGAS inhibitor, named RU.521, had been discovered through high-throughput screening and structural improvement [163]. Immunoblot analysis confirmed that the intraperitoneal injection of RU.521 in mice models targeted the cGAS-STING signaling pathway and reduced signs of colitis in mice. Notably, Ru.521 totally rescued the colitogenic phenotype in mice lacking GSDMD [160]. These data suggest that RU.521 might potentially be useful for protecting against the development of colitis, or even for the treatment of IBD.

Interestingly, as previously explained, also ANP, a peptide hormone, can be used as an inhibitor of cGAS-STING signaling pathway. Indeed, intraperitoneal injection of ANP recombinant protein in DMXAA-treated colitis mice distinctly reversed the increased expression of cGAS and phosphorylation of STING, IRF3 and TBK1, while reducing body loss, colon shortening, DAI score and improving structural injury in a murine model of DSS-induced colitis. Intriguingly, activation of cGAS-STING pathway reduced the expression of ANP and its receptor in the intestine of IBD patients and colitis mice [158,159].

Recently, several novel molecules have been developed for the inhibition of the cGAS-STING-TBK1 signaling pathway as treatment of other autoimmune diseases, such as SLE, RA, Sjogren syndrome or neurodegenerative diseases, in which this pathway appears to play a key role. Such inhibitors reported to decrease the levels of pro-inflammatory cytokines and the inflammatory signaling at the cellular and the animal levels. Indeed, the covalent STING inhibitor compound 36, as well as the compound 33, demonstrated to significantly decrease systemic cytokine responses respectively in mice models treated with the STING agonist and TREX1-/- mice, thereby attenuating symptoms of autoinflammatory diseases in vivo [164].

Particularly, three types of STING inhibitors have been developed: the first including covalent inhibitors that form specific covalent bonds with Cys91, Cys88/91 or His16, of which compound 36 is the representative [165]; the second, such as compound 28, can disrupt STING/TBK1 interactions [166]; and the third, including compound 33, compete with 2′3′-cGAMP at the CDN binding site of STING [167]. Currently, both the first and third types have demonstrated much more potent activity than the second type inhibitor, since the IC50 in at the nanomolar level instead of micromolar level, as second type [164].

Finally, the transcription factor Nrf2 has been recently discovered as a novel inhibitor of STING expression in human but not murine cells, through the destabilization of TMEM173 mRNA. Moreover, the demonstration that treatment with the Nrf2 inducer sulforaphane, or a cell permeable derivative of itaconate reduced STING-dependent release of type I IFNs, promoted the idea that Nrf2 represent a valid therapeutic target for the treatment of STING-dependent diseases [168].

These results, derived from in vitro studies and in animal models, reveal the importance and potential of cGAS-STING signaling inhibitors as a new therapeutic treatment for autoinflammatory diseases, and IBD in particular. However, human studies are required to confirm these promising findings.

### 6.3. Extracellular Vesicles Therapeutics

Orally or intravenously administered macrovesicles and nanoparticles (NPs) can be used as drug carriers in IBD. Notably, in 2001, the targeted accumulation of small nanoparticles, sizes around 100 nm, in the inflamed intestinal mucosa of murine models, compared to healthy colonic mucosa, has been revealed [169,170]. Indeed, NPs can target and accumulating selectivity in intestinal inflammatory sites, due to: (1) the defective mucus layer and the loss of barrier integrity, which promotes NPs infiltration and transcellular transport; (2) the increased mucus secretion, which facilities NPs’ adhesion through hydrophilic functional groups on their surfaces, and diffusion through the mucus layer; (3) the infiltration of immune cells, such as macrophages and neutrophils, which promotes cellular uptake of NPs to the inflammation sites [171,172,173,174,175,176,177,178]. Particularly, several studies revealed that the best tissue-penetrating NPs for the treatment of IBD patients are less than 200 nm in size and have a negative surface charge, such as anionic liposomes, in order to benefit from the enhanced permeability effect of inflammation and from the accumulation of positively charged proteins at the damaged epithelium of IBD patients [179].

As for the ligand/receptor-dominated IEC targeting, some membrane upregulated proteins in IBD patients, such as the glycoprotein CD98, the peptide transporter 1 or other transports could be used as anchors for the attachment of NPs [180,181,182,183,184]. Moreover, ligands and adhesion molecules involved in leukocyte trafficking have recently become a popular approach for drug delivery. Indeed, NPs mimicking the structures of these ligand are being designed to attach to endothelial cells and release their content to the adjacent immune cells. Among them, synthesized NPs with recombinant P-selectin glycoprotein ligand-1 (PSGL-1) conjugated to PEGylated PLA particles showed a significantly stronger ability to adhere to the inflamed endothelium in an in vivo model, compared to unconjugated PEG-PLA particles [185,186]. Another emerging approach is targeting activated macrophages via recognition of their overexpressed receptors, by NPs -based cell-specific therapy in IBD. Indeed, different studies used mannosylated poly-(amidoamine)-based NPs to target mannose and galactosylated chitosan NPs to target the macrophage galactose/N-acetyl galactosamine-specific lectins. Thus, nanoparticle-based drug delivery can not only deliver natural compounds and conventional agents, such as corticosteroids, immunosuppressive and biological drugs, to the inflammatory sites of the intestinal mucosa, but also increase local bioavailability and concentrations of the drug, while reducing exposure in healthy tissue [187,188,189]. Hence, these delivery systems would be able to facilitate remission in IBD patients by enhancing treatment efficacy and reducing side effects and systemic toxicity [190].

NPs can pass through different physiological barriers through their low immunogenicity and precise targeting, loading several therapeutic agents, such as proteins, cfDNA, RNA or anti-sense oligonucleotides, which could achieve high stability by NPs’ protection away from DNase’ degradation [191].

### 6.4. Oligonucleotide Therapeutics

Therapeutic oligonucleotides are synthesized nucleic acids which interferes with the pathogenesis of IBD. Interestingly, multiple forms of oligonucleotides are being applied in medicine, entailing different molecular mechanisms, among which the inhibition of the translational process of messenger ribonucleic acid (mRNA) transcripts is the most studied for IBD treatment.

Indeed, antisense oligonucleotide (ASO) or synthetic oligonucleotide-based therapy represent a promising approach for the management of IBD, including small-interfering RNA (siRNA), microRNA, aptamers and ASOs.

In detail, siRNAs are double-stranded RNA which can suppress the expression of a gene through a process called RNA interference (RNAi) [192]. Aptamers are single-stranded RNA or DNA synthetic oligonucleotides (size 20–80 pb), selected by a process referred to as SELEX (systematic evolution of ligands by exponential enrichment) [193,194].

ASOs are short single-stranded polymers of nucleic acids (DNA or RNA), designed to interact specifically with target mRNA [195]. Accordingly, data from preclinical work and clinical trials are accumulating for several oligonucleotide compounds, including alicaforsen, DIMS0150 and BL-7040, Mongersen, STNM01, hgd40, ASA targeting NF-κBp65. As previously showed, DIMS0150 and BL-7040 are two oligonucleotides that enhance the activity of TLR9.

Alicaforsen (ISIS 2302) is a human RNase H-dependent, 20 base-long, phosphorothioate ASO that blocks, by hybridization to the mRNA, the production of the intercellular adhesion molecule [ICAM]-1, a trans- membrane glycoprotein that regulates rolling and adhesion of leukocytes to the inflamed intestinal mucosa [196,197]. While data in patients with Crohn’s Disease regarding intravenous administration are inconsistent [198,199,200,201], alicaforsen administration by enema showed promising results in patients with left-side and distal ulcerative colitis or chronic pouchitis [202,203].

Mongersen (GED0301) is a synthetic phosphorothioate single-stranded DNA oligonucleotide that matches the region 107–128 of the human SMAD7 complementary DNA sequence [204]. By inhibiting SMAD7, it restores the transforming growth factor- β1 (TGF- β1)/SMAD2/3 associated anti-inflammatory signaling [205]. A multicenter phase II clinical trial revealed that orally administered Morgensen was superior to placebo in the treatment of patients with active steroid-dependent/resistant CD [206]. However, a phase III clinical trial was discontinued in advance, due to lack of efficacy.

STNM01 is a double-stranded RNA oligonucleotide silencing carbohydrate sulfotransferase 15 (CHST15), an enzyme involved in fibrogenic processes [207]. The submucosal injection of STNM01 during colonoscopy showed higher rates of mucosal healing, clinical remission and lower histological scores compared to placebo in CD and UC patients [207,208].

Hgd40 is a specific DNAzyme that efficiently cleaves and inhibits GATA3 RNA expression, a transcription factor which regulates the commitment of naïve T cells towards the type 2 T helper (Th2) phenotype [209,210]. Mice given hdg40 intra-rectally were protected from development of colitis and expressed lower level of proinflammatory cytokines than control DNAzyme [210].

Furthermore, given the role of p65 subunit of NF-κB in the development of colitis, an ASO targeting p65 has been produced. The intravenous, oral or intracolonic administration of ASO targeting p65 reduced the mucosal inflammation in murine models of TNBS- and DSS-induced colitis [211,212] and the proinflammatory cytokine expression by macrophages of the intestinal mucosa of IBD patients [212]. Nonetheless, clinical studies assessing the safety profile and efficacy of NF-κBp65 ASO are still lacking.

These and other promising novel oligonucleotides for the treatment of IBD are presented in Table 2.

### 6.5. eccDNA Therapeutics

As previously explained, eccDNA appears to correlate with the severity of intestinal inflammation in patients with IBD. Moreover, several studies have now confirmed that eccDNA, and especially, the larger circular DNA fragments actively contribute to the promotion of carcinogenesis, greater tumor heterogeneity, malignant phenotype of cancer cells and development of drug resistance genes, thus promoting the evolution of neoplasia and worsening the prognosis of patients [73]. Given these premises, therefore, it is possible to hypothesize an active role of eccDNA in the promotion of intestinal inflammation and possibly in the evolution to low-grade dysplasia, high-grade dysplasia and finally cancer-associated with colitis (CAC), that IBD patients may develop. Indeed, eccDNA could be the site of formation and amplification of oncogenes, thus contributing to the development of a malignant phenotype by intestinal epithelial cells. In addition, similar to what occurs in cancer, circular DNA could harbor resistance genes to drugs commonly used in IBD, such as immunosuppressants, biologic drugs, or small molecules, thereby causing a primary nonresponse or secondary loss of response to therapy and contributing to changes in the immunologic environment over the course of disease evolution. Given all these findings, it is reasonable that targeting the eccDNA would be a good therapeutic strategy in IBD patients, and especially in patients who have developed low- or high-grade dysplasia or even CAC. Different studies have demonstrated that the total amount of eccDNA harboring oncogenes can be reduced by drug treatment in cells [219,220,221,222]. Particularly, a study involving 16 patients with advanced ovarian cancer, treated with a non-cytotoxic dose of hydroxyurea showed a decrease in the number of metaphase spreads containing eccDNA in cancer cells [223]. Next to drug treatment, also radiotherapy, using ionizing radiation proved to significantly reduce the number of eccDNA carrying MYC and MDR1 genes in human epidermoid and colon carcinoma cell lines [224,225]. Finally, CRISPR/Cas9 may represent a novel and promising methods to guide the cleavage of targeted eccDNA [226].

On the other hand, given the higher stability, the longer half-life than their linear counterpart, the resistivity to exonuclease digestion due to their circular structure, eccDNAs might constitute the most needed molecules to deliver required coding genes or regulatory RNAs for the treatment of IBD.

All of the molecular therapies mentioned above are depicted in Figure 1.

## 7. Concluding Remarks and Future Perspectives

In this review, we performed a comprehensive overview of the role of cell-free DNA and circular DNA in inflammatory bowel diseases, ranging from potential diagnostic to therapeutic applications. As we reviewed above, different types of cfDNAs can be detected in different body fluids, such as blood or feces, potentially representing a novel noninvasive and easily accessible biomarker. Rising from “passive” processes of cell death (necrosis, mechanical damage, etc.), “active” processes of cell death (apoptosis, NETosis), defects in DNA repair mechanisms, oxidative stress and inflammatory environment, especially for eccDNA, and processes of cell-to-cell communications (microvescicles, exosomes, nanovesicles), the concentration of cfDNA seems to be associated with the severity of disease in IBD patients and murine models of colitis.

In each process, cfDNAs represent detectable signals of the processes which are ongoing in the organism, and of its own reaction, thereby, giving researchers the possibility of both passively monitoring or actively controlling these processes. Growing evidence shows that cfDNAs play important roles in several inflammatory diseases, including IBD, most of whom are still unknown, thus requiring further research. The creation of in vitro models would allow us to better understand the exact meaning of these processes. Notably, Moller et al. created a dual-fluorescence biosensor cassette, which, upon the delivery of pairs of CRISPR/Cas9 guide RNAs, is able to generate in human cells eccDNA from intergenic and genic loci, of different sizes, thus allowing researchers to study the cellular impact, persistence and function of eccDNAs in different tissues, such as human intestinal epithelium [71].

Furthermore, one goal for the future is the focus on the rapid progression of single cell-based studies on immune system disorders, autoinflammatory or autoimmune diseases. In the past decade, single-cell RNA sequencing (scRNA-seq) has driven a true revolution in the field of immunological research, since it allows for a better understanding of the heterogeneity associated with individual immune cells and immunological responses at the molecular level both under physiological and pathological conditions [227]. The development of similar technologies for the purification and sequencing of cfDNA, and especially eccDNAs will open new roads to translational research in the field of immunology and cancer. Additionally, there is an urgent need to innovate more bioinformatics tools for data analysis and integration from different technologies and data types, in order to standardize the immune cells annotation and to utilize the whole spectrum of available data.

Therefore, cfDNA may represent the much-needed noninvasive biomarker, from liquid biopsy, for early diagnosis, disease monitoring, prediction of drug response, and early detection of progression to dysplasia in IBD.

Herein, we reviewed the current state of art regarding the development of molecular therapies in IBD, with attention to the role played by cGAS-STING signaling, TLR9 signaling, and discussing several other potential oligonucleotide-based therapies in IBD. We also discussed the rationale of employing NPs for drug delivery and provided novel interesting insights into the use of different molecules, such as eccDNA for the treatment of IBD.

However, further research is needed to better characterize the origins, functions, and biological features of cfDNA and eccDNA, which could contribute to the elucidation of IBD pathogenesis and to the development of novel molecular therapeutic strategies.

## Figures and Tables

**Figure 1 cells-12-01953-f001:**
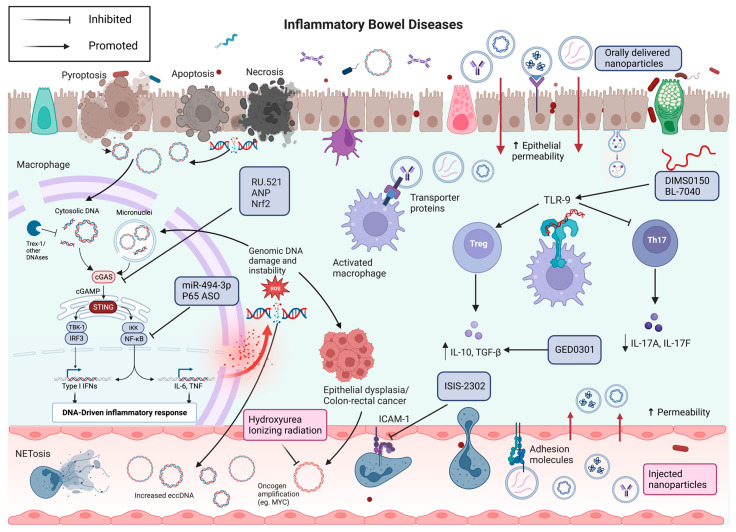
From pathogenesis to molecular therapies for IBD. The normal intestinal epithelium is characterized by the production and release of low amounts of cfDNA, present both locally and at a distance. In IBD, the increased rates of cellular turnover and the inflammation lead to increased production and release of cfDNA, both in free-form and within microvesicles. dsDNA is usually disrupted by Trex1, but Trex1 loss or malfunction can lead to the activation of the cGAS surveillance present in the cytoplasm. Moreover, chromosome damage, due to the involvement of centromeres, can cause chromatids mis-segregation in the two daughter cells, leading to the formation of micronuclei. When the micronuclear envelope ruptures, the DNA is exposed to cGAS surveillance. When cGAS binds to free cytoplasmatic DNA, it synthesizes cGAMP, which activates STING on the endoplasmic reticulum surface. STING, in turn, through the transcription factors IRF3 and NF-kB induces the production of pro-inflammatory cytokines. RU.521, ANP and Nrf2 inhibits the cGAS-STING pathway, thus reducing inflammation. P65 ASO and miR-494-3p inhibit NF-kB and the consequent production of inflammatory cytokines. Conversely, cfDNA can also activate TLR9 resulting in an immunomodulatory effect, depending on the specific DNA sequence. The TLR9 agonists DIMS0150 and BL-7040 ameliorates intestinal inflammation by inducing regulatory T cells and anti-inflammatory cytokines, and reducing pro-inflammatory cytokines, such as il-17A and IL-17F. ISIS-2303 blocks leukocytes trafficking, by inhibiting the production of ICAM-1. Persistent inflammation and continued DNA damage may lead to the accumulation of DNA mutations and amplification of oncogenes on eccDNA, thereby causing the development of dysplasia, and eventually colorectal cancer. In this scenario, the use of low doses of hydroxyurea or ionizing radiation could destroy those fragments of eccDNA. Finally, both orally and intravenously administrated nanovesicles can represent the optimal delivery system of several molecules towards sites of intestinal inflammation. Created with BioRender.com (accessed on 17 June 2023).

**Table 1 cells-12-01953-t001:** The role of cfDNA as a diagnostic biomarker in IBD.

Molecule	Disease	Sample Type	Study Design	Methods	Outcomes	Ref.
Exosomal mtDNA and nDNA	Murine colitis	Murine colon lavage and plasma	Preclinical/clinical	Quantitative polymerase chain reactions (qPCR) targeting Hist1h3F and mtCOI	Levels of exosomal nDNA, mtDNA from colon lavage and plasma of DSS mice positively correlated with disease activity	[101]
	Human CD	Human plasma			Levels of exosomal nDNA and mtDNA in human plasma positively correlated with disease activity	
mtDNA, nDNA and NETosis	Dss murine colitis	Plasma and colonic samples	Preclinical	ecDNA: Real-time PCR on the Mastercycler realplex	Plasma and colonic ecDNA concentration increased over time, showing correlation with cells undergoing NETosis	[28]
				NETs: flow cytometer and analyzed by FCSExpress 6.0 software		
Nuclear and mitochondrial cfDNA	CD and UC patients in remission	Plasma	Clinical	Quantitative PCR	Increased mt/nDNA as compared to controls	[60]
cfDNA and methylation levels	Murine DSS colitis	Plasma	Preclinical	DNA methylation sensitive restriction enzyme Hpa II or BstU I to test DNA methylation, PCR	Increased levels of circulating cfDNA and reduced methylation levels in colitis and colitis associated colon cancer	[110]
cfDNA and NETs	Murine DSS colitis and UC patients	Colonic samples (mice), plasma (UC)	Preclinical/clinical	Multiphoton surgical microscopy (colon), PCR (plasma)	Increased levels of circulating cfDNA in both DSS-mice colon samples and plasma compared to controls, and in plasma of UC patients	[111]
eccDNA	UC and CD patients	Colonic mucosal samples	Monocentric observational case-control study	Circle finder	Increased eccDNA levels in IBD as compared to healthy controls	Accepted abstract at UEG 2023

cfDNA cell-free DNA, nDNA nuclear DNA, eccDNA extrachromosomal circular DNA, mtDNA, mitochondrial DNA, ecDNA extrachromosomal DNA, UC Ulcerative colitis, CD Crohn’s Disease, DSS dextran sodium sulfate, PCR Polymerase Chain Reaction.

**Table 2 cells-12-01953-t002:** Oligonucleotide-based therapy in IBD.

Molecule Name	Compound	Target	Mechanism of Action	Disease	Study Design—References	Outcomes	Route of Administration	Developmental Stage
Mongersen	ASO	SMAD7	Restoring TGF-β1 activity	CD	Randomized quadruple blind, clinical trial [213]	Not superior to placebo	Oral	Phase III
Alicaforsen	ASO	ICAM-1	Leukocyte trafficking	CD	Randomized double blind clinical trial [NCT00048113]	Ongoing	Intravenous	Phase III
				UC	Double-blind, placebo controlled clinical trial [214]	Statistical benefit over placebo for prolonged reduction in DAI	Enema	Phase II
				Pouchitis	Double-blind randomized controlled clinical trial [NCT02525523]	Ongoing	Enema	Phase III
STNM01	ds-RNA	CHST15	Inhibition of collagen fibril formation	CD	Randomized, double blind, placebo controlled clinical trial [208]	Amelioration of SES-CD and fibrosis	Intracolonic	Phase I
				UC	Randomized, multicenter, double-blind, placebo-controlled clinical trial	Higher rates of mucosal healing and clinical remission in left-sided refractory colitis	Intracolonic	Phase IIa
Hdg40/SB012	DNAzyme	GATA3	Inhibition of Th2-driven response	UC	Double blind randomized clinical trial [NCT02129439]	Clinical and endoscopic improvement of disease activity	Intrarectal	Phase IIa
DIMS0150, Cobitolimod	ss-DNA	TLR9	Induction of anti-inflammatory cytokines	UC	Randomized quadruple blind placebo controlled clinical trial [NCT01493960]	Higher clinical remission in moderate-to-severe UC	Intracolonic	Phase III
BL-7040, Monarsen	Synthetic oligonucleotide	TLR9	Induction of anti-inflammatory cytokines	UC	Single group assignment, open label clinical trial [143]	Higher clinical response and remission in moderate UC	Oral	Phase II
P65 ASO	ASO	nF-kB	Reduction in pro-inflammatory cytokines	UC	Murine models [211]	Downregulation of NF-kB and proinflammatory cytokines	Intrarectal	Pre-clinical studies
miR-494-3p	microRNA	IKKβ/NF-κB, EDA2R/EDA-A2	inhibits M1 macrophage recruitment, suppresses colonic stemness and epithelial repair	DSS induced colitis in mice	Murine models [215]	Ameliorated severity of colonic colitis	intraperitoneal injection	Pre-clinical studies
interleukin-10 (IL-10) mRNA	mRNA		anti-inflammatory cytokine	DSS induced colitis in mice	Murine models [216]	Anti-inflammatory effect on intestinal mucosa		Pre-clinical studies
AMT-101	Chx386–hIL-10 fusion protein	IL-10 receptor	exerting IL-10 anti-inflammatory activity	DSS induced colitis in mice	Murine models [217]	Efficient transcytosis towards intestinal lamina propria and activation of IL-10R	Oral	Pre-clinical studies
AptMincleDRBL	Aptamer	Mincle	blocking Mincle (PRR) pathway	DSS induced colitis in mice	Murine models [218]	Reduction in disease activity	intraperitoneal injection	Pre-clinical studies

ASO antisense oligonucleotide, CD Crohn’s disease, UC ulcerative colitis, DSS dextran sodium sulfate, IL Inteleukin, CHST15 carbohydrate sulfotransferase 15, ds double-stranded, ICAM intercellular adhesion molecule, NF-kB nuclear factor kappa-light-chain-enhancer of activated B cells, ss single-stranded, TGF transforming growth factor, mRNA Messenger ribonucleic acid, Th2 type 2 T helper, TLR Toll-like receptor, PRR pattern recognition receptors.

## Data Availability

No new data were created or analyzed in this study. Data sharing is not applicable to this article.

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
