# Peer review of "Circular and Circulating DNA in Inflammatory Bowel Disease: From Pathogenesis to Potential Molecular Therapies"

_cells, 2023, doi:10.3390/cells12151953_

Round 1

Reviewer 1 Report

Overall a comprehensive review on cell-free DNA (cfDNA) in inflammatory bowel disease and how this form of DNA might be used both in the diagnosis and as a potential therapy for this condition. Table 1 lists a range of oligonucleotides that have potential in therapy and a single figure is presented to represent signaling in IBD and where therapy might be applied.

Specific comments

1. The review is generally well structured outlining the different cfDNAs that are implicated.

2. Separate sections cover the main players.

3. The section on role of computational biology is poorly organised and describes little io computational methods used and the major findings as they relate to diagnosis. In fact it seems to question the use of cfDNA rather than support it.

4. Use of english poor and with misspellings e.g. findable line 96;shedded line 111;bounded line 112; here's line 115 ;in site line 127;wether line 218 ; wether line 223 ; trough line 232.

5. Use of definite artice  also the clearence line 113 ;etc The english needs to be addressed throughout. These are just a few examples.

6. Would help to have a Table to describe some of the approaches to diagnosis similar to therapy

4. 

Needs to be addressed throughout. Poor.

Author Response

Comments and Suggestions for Authors:

Reviewer 1

Overall a comprehensive review on cell-free DNA (cfDNA) in inflammatory bowel disease and how this form of DNA might be used both in the diagnosis and as a potential therapy for this condition. Table 1 lists a range of oligonucleotides that have potential in therapy and a single figure is presented to represent signaling in IBD and where therapy might be applied.

Specific comments

  1. The review is generally well structured outlining the different cfDNAs that are implicated. Thanks
  2. Separate sections cover the main players.Thanks
  3. The section on role of computational biology is poorly organised and describes little io computational methods used and the major findings as they relate to diagnosis. In fact it seems to question the use of cfDNA rather than support it.

Response: We thank the reviewer for the valuable comment. We have revised the section of computational biology and tracked the changes in the revised manuscript. Unlike linear cell free DNA (cf DNA),cell free extrachromosomal circular DNA (cf-eccDNA) has not been studied and used in diagnosis of IBD yet. This section reflects our view and experience how cf-eccDNA can be used to discriminate between two disease conditions, or disease from control. The field of establishing eccDNA biomarkers is new and we as part of a project that explores the new opportunities circular DNA creates in early diagnosis, screening and monitoring of disease, have gained some insights on the potential and current limitations of the use of cf-eccDNA in disease diagnosis that we shared in this section. 

  1. Use of english poor and with misspellings e.g. findable line 96;shedded line 111;bounded line 112; here's line 115 ;in site line 127;wether line 218 ; wether line 223 ; trough line 232.

Response: We apologize with the reviewer for these inaccuracies. We corrected these mistakes and took the opportunity to improve the quality of the English throughout the entire manuscript.

  1. Use of definite artice  also the clearence line 113 ;etc The english needs to be addressed throughout. These are just a few examples.

Response: We thank the reviewer for the comment. We improved the quality of the English throughout the entire manuscript.

  1. Would help to have a Table to describe some of the approaches to diagnosis similar to therapy

Response: We thank the reviewer for the interesting suggestion. We added a table in the manuscript to describe the potential diagnostic approaches.

Finally, as asked by the editor, we added a graphical abstract in the manuscript.

Reviewer 2 Report

The authors of the article have provided a very comprehensive, detailed, and well-structured summary of the pathogenetic, diagnostic, and therapeutic applications of cell-free DNA and circular DNA in IBD.
The topic is very interesting and forward-looking. The authors analyze the biological impact of cfDNA and cicrDNA in the light of IBD aspects, both pro- and contra- analyses, and their potential pathobiological effects based on the results of model experiments performed so far. Their role in diagnosis is described in detail, and in addition, broad and novel aspects of therapeutic options are presented.
A few minor typos and stylistic corrections are needed, e.g.,
Line 123: INF-alpha (IFN);

Line 171: Gran positive and Gram negative (bacteria);

Line 205: occationally (occasionally) expresse (d);

Lines 218-219: wether (whether), the end of the sentence is missing. 

The paragraph between lines 176-187 does not relate in detail to the topic and should be shortened or omitted. 

After making these minor corrections, I recommend that the article be accepted for publication.

This article, if accepted for publication, is also eligible for inclusion in the Topic "Cancer Stem Cells, DNA Methylation, and DNA Sequences: Their Diagnostic and Therapeutic Applications". 

Minor English polishing is required.

Author Response

Comments and Suggestions for Authors:

Reviewer 2

The authors of the article have provided a very comprehensive, detailed, and well-structured summary of the pathogenetic, diagnostic, and therapeutic applications of cell-free DNA and circular DNA in IBD.
The topic is very interesting and forward-looking. The authors analyze the biological impact of cfDNA and cicrDNA in the light of IBD aspects, both pro- and contra- analyses, and their potential pathobiological effects based on the results of model experiments performed so far. Their role in diagnosis is described in detail, and in addition, broad and novel aspects of therapeutic options are presented.
1. A few minor typos and stylistic corrections are needed, e.g.,
Line 123: INF-alpha (IFN);

Line 171: Gran positive and Gram negative (bacteria);

Line 205: occationally (occasionally) expresse (d);

Lines 218-219: wether (whether), the end of the sentence is missing. 

Response: We thank the reviewer for pointing out these inaccuracies and typos. We corrected typos throughout the whole text.

  1. The paragraph between lines 176-187 does not relate in detail to the topic and should be shortened or omitted. 

Response: We thank the reviewer for the valuable comment which we share. As you suggested, we shorted this paragraph in the manuscript.

After making these minor corrections, I recommend that the article be accepted for publication.

This article, if accepted for publication, is also eligible for inclusion in the Topic "Cancer Stem Cells, DNA Methylation, and DNA Sequences: Their Diagnostic and Therapeutic Applications".

Finally, as asked by the editor, we added a graphical abstract in the manuscript.
